# Residual Scheduling: A New Reinforcement Learning Approach to Solving Job Shop Scheduling Problem

## Abstract

Job-shop scheduling problem (JSP) is a mathematical optimization problem widely used in industries like manufacturing, and flexible JSP (FJSP) is also a common variant. Since they are NP-hard, it is intractable to find the optimal solution for all cases within reasonable times. Thus, it becomes important to develop efficient heuristics to solve JSP/FJSP. A kind of method of solving scheduling problems is construction heuristics, which constructs scheduling solutions via heuristics. Recently, many methods for construction heuristics leverage deep reinforcement learning (DRL) with graph neural networks (GNN). In this paper, we propose a new approach, named residual scheduling, to solving JSP/FJSP. In this new approach, we remove irrelevant machines and jobs such as those finished, such that the states include the remaining (or relevant) machines and jobs only. Our experiments show that our approach reaches state-of-the-art (SOTA) among all known construction heuristics on most well-known open JSP and FJSP benchmarks. In addition, we also observe that even though our model is trained for scheduling problems of smaller sizes, our method still performs well for scheduling problems of large sizes. Interestingly in our experiments, our approach even reaches zero gap for 49 among 50 JSP instances whose job numbers are more than 150 on 20 machines.

## 1 Introduction

The *job-shop scheduling problem* (*JSP*) is a mathematical optimization (MO) problem widely used in many industries, like manufacturing (Zhang et al., 2020; Waschneck et al., 2016). For example, a semiconductor manufacturing process can be viewed as a complex JSP problem (Waschneck et al., 2016), where a set of given jobs are assigned to a set of machines under some constraints to achieve some expected goals such as minimizing makespan which is focused on in this paper. While there are many variants of JSP (Abdolrazzagh-Nezhad and Abdullah, 2017), we also consider an extension called *flexible JSP* (*FJSP*) where job operations can be done on designated machines.

A generic approach to solving MO problems is to use mathematical programming, such as mixed integer linear programming (MILP) and constraint satisfaction problem (CSP). Two popular generic MO solvers for solving MO are *OR-Tools* (Perron and Furnon, 2019) and *IBM ILOG CPLEX Optimizer* (abbr. *CPLEX*) (Cplex, 2009). However, both JSP and FJSP, as well as many other MO problems, have been shown to be NP-hard (Garey and Johnson, 1979; Lageweg et al., 1977). That said, it is unrealistic and intractable to find the optimal solution for all cases within reasonable times. These tools can obtain the optimal solutions if sufficient time (or unlimited time) is given; otherwise, return best-effort solutions during the limited time, which usually have gaps to the optimum. When problems are scaled up, the gaps usually grow significantly.

In practice, some heuristics (Gupta and Sivakumar, 2006; Haupt, 1989) or approximate methods (Jansen et al., 2000) were used to cope with the issue of intractability. A simple greedy approach is to

use the heuristics following the so-called *priority dispatching rule* (*PDR*) (Haupt, 1989) to construct
solutions. These can also be viewed as a kind of *solution construction heuristics* or *construction
heuristics*. Some of PDR examples are *First In First Out* (*FIFO*), *Shortest Processing Time* (*SPT*),
*Most WorK Remaining* (*MWKR*), and *Most Operation Remaining* (*MOR*). Although these heuristics
are usually computationally fast, it is hard to design generally effective rules to minimize the gap to
the optimum, and the derived results are usually far from the optimum.

Furthermore, a generic approach to automating the design of heuristics is called *metaheuristics*, such
as tabu search (Dell'Amico and Trubian, 1993; Saidi-Mehrabad and Fattahi, 2007) , genetic algorithm
(GA) (Pezzella et al., 2008; Ren and Wang, 2012), and PSO algorithms (Lian et al., 2006; Liu et al.,
2011). However, metaheuristics still take a high computation time, and it is not ensured to obtain the
optimal solution either.

Recently, deep reinforcement learning (DRL) has made several significant successes for some
applications, such as AlphaGo (Silver et al., 2016), AlphaStar (Vinyals et al., 2019), AlphaTensor
(Fawzi et al., 2022), and thus it also attracted much attention in the MO problems, including chip
design (Mirhoseini et al., 2021) and scheduling problems (Zhang et al., 2023). In the past, several
researchers used DRL methods as construction heuristics, and their methods did improve scheduling
performance, illustrated as follows. Park et al. (2020) proposed a method based on DQN (Mnih et al.,
2015) for JSP in semiconductor manufacturing and showed that their DQN model outperformed GA
in terms of both scheduling performance (namely gap to the optimum on makespan) and computation
time. Lin et al. (2019) and Luo (2020) proposed different DQN models to decide the scheduling action
among the heuristic rules and improved the makespan and the tardiness over PDRs, respectively.

A recent DRL-based approach to solving JSP/FJSP problems is to leverage graph neural networks
(GNN) to design a size-agnostic representation (Zhang et al., 2020; Park et al., 2021b,a; Song et al.,
2023). In this approach, graph representation has better generalization ability in larger instances
and provides a holistic view of scheduling states. Zhang et al. (2020) proposed a DRL method
with disjunctive graph representation for JSP, called *L2D* (*Learning to Dispatch*), and used GNN
to encode the graph for scheduling decision. Besides, Song et al. (2023) extended their methods
to FJSP. Park et al. (2021b) used a similar strategy of (Zhang et al., 2020) but with different state
features and model structure. Park et al. (2021a) proposed a new approach to solving JSP, called
*ScheduleNet*, by using a different graph representation and a DRL model with the graph attention for
scheduling decision. Most of the experiments above showed that their models trained from small
instances still worked reasonably well for large test instances, and generally better than PDRs. Among
these methods, ScheduleNet achieved state-of-the-art (SOTA) performance. There are still other
DRL-based approaches to solving JSP/FJSP problems, but not construction heuristics. Zhang et al.
(2022) proposes another approach, called Learning to Search (L2S), a kind of search-based heuristics.

In this paper, we propose a new approach to solving JSP/FJSP, a kind of construction heuristics, also
based on GNN. In this new approach, we remove irrelevant machines and jobs, such as those finished,
such that the states include the remaining machines and jobs only. This approach is named *residual
scheduling* in this paper to indicate to work on the remaining graph.

Without irrelevant information, our experiments show that our approach reaches SOTA by outper-
forming the above mentioned construction methods on some well-known open benchmarks, seven
for JSP and two for FJSP, as described in Section 4. We also observe that even though our model
is trained for scheduling problems of smaller sizes, our method still performs well for scheduling
problems of large sizes. Interestingly in our experiments, our approach even reaches zero gap for 49
among 50 JSP instances whose job numbers are more than 150 on 20 machines.

## 2 Problem Formulation

### 2.1 JSP and FJSP

A $n \times m$ JSP instance contains $n$ jobs and $m$ machines. Each job $J_j$ consists of a sequence of $k_j$
operations $\{O_{j,1}, \ldots, O_{j,k_j}\}$, where operation $O_{j,i}$ must be started after $O_{j,i-1}$ is finished. One
machine can process at most one operation at a time, and preemption is not allowed upon processing
operations. In JSP, one operation $O_{j,i}$ is allowed to be processed on one designated machine, denoted
by $M_{j,i}$, with a processing time, denoted by $T_{j,i}^{(op)}$. Table 1 (a) illustrates a $3 \times 3$ JSP instance, where
the three jobs have 3, 3, 2 operations respectively, each of which is designated to be processed on

one of the three machines $\{M_1, M_2, M_3\}$ in the table. A solution of a JSP instance is to dispatch all operations $O_{j,i}$ to the corresponding machine $M_{j,i}$ at time $\tau_{j,i}^{(s)}$, such that the above constraints are satisfied. Two solutions of the above 3x3 JSP instance are given in Figure 1 (a) and (b).

Table 1: JSP and FJSP instances

<table>
<tr><td colspan="5">(a) A $3 \times 3$ JSP instance</td><td colspan="5">(b) A $3 \times 3$ FJSP instance</td></tr>
<tr><td>Job</td><td>Operation</td><td>$M_1$</td><td>$M_2$</td><td>$M_3$</td><td>Job</td><td>Operation</td><td>$M_1$</td><td>$M_2$</td><td>$M_3$</td></tr>
<tr><td rowspan="3">Job 1</td><td>$O_{1,1}$</td><td>3</td><td></td><td></td><td rowspan="3">Job 1</td><td>$O_{1,1}$</td><td>3</td><td>2</td><td></td></tr>
<tr><td>$O_{1,2}$</td><td></td><td></td><td>5</td><td>$O_{1,2}$</td><td>3</td><td></td><td>5</td></tr>
<tr><td>$O_{1,3}$</td><td></td><td>4</td><td></td><td>$O_{1,3}$</td><td></td><td>4</td><td>3</td></tr>
<tr><td rowspan="3">Job 2</td><td>$O_{2,1}$</td><td></td><td></td><td>2</td><td rowspan="3">Job 2</td><td>$O_{2,1}$</td><td></td><td></td><td>2</td></tr>
<tr><td>$O_{2,2}$</td><td></td><td>4</td><td></td><td>$O_{2,2}$</td><td></td><td>4</td><td></td></tr>
<tr><td>$O_{2,3}$</td><td>3</td><td></td><td></td><td>$O_{2,3}$</td><td>3</td><td></td><td></td></tr>
<tr><td rowspan="2">Job 3</td><td>$O_{3,1}$</td><td>3</td><td></td><td></td><td rowspan="2">Job 3</td><td>$O_{3,1}$</td><td>3</td><td>4</td><td></td></tr>
<tr><td>$O_{3,2}$</td><td></td><td></td><td>2</td><td>$O_{3,2}$</td><td>2</td><td></td><td>2</td></tr>
</table>

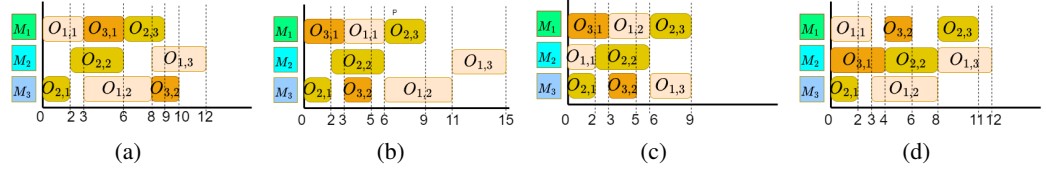

(a)     (b)     (c)     (d)

Figure 1: Both (a) and (b) are solutions of the 3x3 JSP instance in Table 1 (a), and the former has the minimal makespan, 12. Both (c) and (d) are solutions of the 3x3 FJSP instance in Table 1 (b), and the former has the minimal makespan, 9.

While there are different expected goals, such as makespan, tardiness, etc., this paper focuses on makespan. Let the first operation start at time $\tau = 0$ in a JSP solution initially. The makespan of the solution is defined to be $T^{(mksp)} = \max(\tau_{j,i}^{(c)})$ for all operations $O_{j,i}$, where $\tau_{j,i}^{(c)} = \tau_{j,i}^{(s)} + T_{j,i}^{(op)}$ denotes the completion time of $O_{j,i}$. The makespans for the two solutions illustrated in Figure 1 (a) and (b) are 12 and 15 respectively. The objective is to derive a solution that minimizes the makespan $T^{(mksp)}$, and the solution of Figure 1 (a) reaches the optimal.

A $n \times m$ FJSP instance is also a $n \times m$ JSP instance with the following difference. In FJSP, all operations $O_{j,i}$ are allowed to be dispatched to multiple designated machines with designated processing times. Table 1 (b) illustrates a $3 \times 3$ FJSP instance, where multiple machines can be designated to be processed for one operation. Figure 1 (c) illustrates a solution of an FJSP instance, which takes a shorter time than that in Figure 1 (d).

## 2.2 Construction Heuristics

An approach to solving these scheduling problems is to construct solutions step by step in a greedy manner, and the heuristics based on this approach is called *construction heuristics* in this paper. In the approach of construction heuristics, a scheduling solution is constructed through a sequence of partial solutions in a chronicle order of dispatching operations step by step, defined as follows. The $t$-th partial solution $S_t$ associates with a *dispatching time* $\tau_t$ and includes a partial set of operations that have been dispatched by $\tau_t$ (inclusive) while satisfying the above JSP constraints, and all the remaining operations must be dispatched after $\tau_t$ (inclusive). The whole construction starts with $S_0$ where none of operations have been dispatched and the dispatching time is $\tau_0 = 0$. For each $S_t$, a set of operations to be chosen for dispatching form a set of pairs of $(M, O)$, called *candidates* $C_t$, where operations $O$ are allowed to be dispatched on machines $M$ at $\tau_t$. An agent (or a heuristic algorithm) chooses one from candidates $C_t$ for dispatching, and transits the partial solution to the next $S_{t+1}$. If there exists no operations for dispatching, the whole solution construction process is done and the partial solution is a solution, since no further operations are to be dispatched.

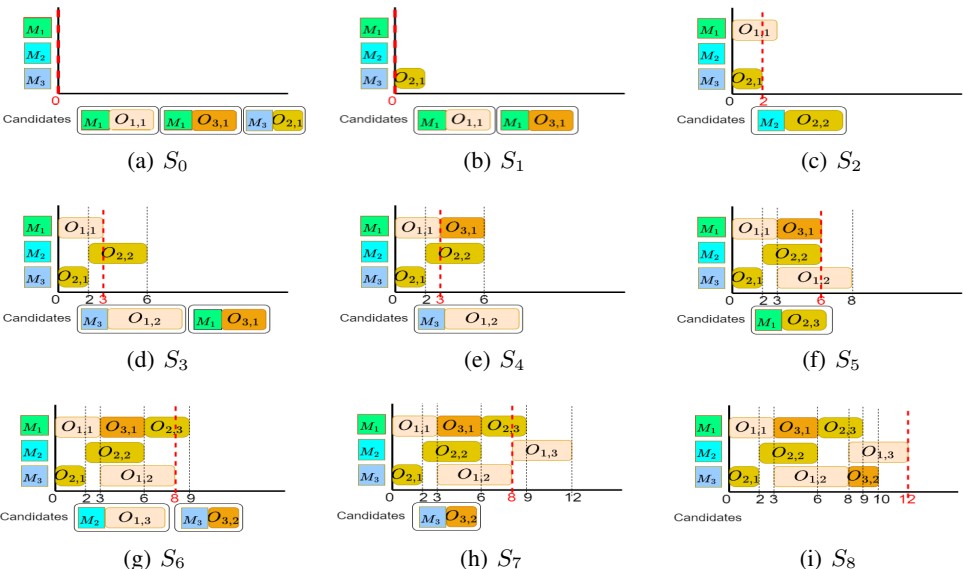

Figure 2: Solution construction, a sequence of partial solutions from $S_0$ to $S_8$.

Figure 2 illustrates a solution construction process for the 3x3 JSP instance in Table 1(a), constructed through nine partial solutions step by step. The initial partial solution $S_0$ starts without any operations dispatched as in Figure 2 (a). The initial candidates $C_0$ are $\{(M_1, O_{1,1}), (M_3, O_{2,1}), (M_1, O_{3,1})\}$. Following some heuristic, construct a solution from partial solution $S_0$ to $S_9$ step by step as in the Figure, where the dashed line in red indicate the time $\tau_t$. The last one $S_9$, the same as the one in Figure 1 (a), is a solution, since all operations have been dispatched, and the last operation ends at time 12, the makespan of the solution.

For FJSP, the process of solution construction is almost the same except for that one operation have multiple choices from candidates. Besides, an approach based on solution construction can be also viewed as the so-called *Markov decision process* (*MDP*), and the MDP formulation for solution construction is described in more detail in the appendix.

## 3 Our Approach

In this section, we present a new approach, called *residual scheduling*, to solving scheduling problems. We introduce the residual scheduling in Subsection 3.1, describe the design of the graph representation in Subsection 3.2, propose a model architecture based on graph neural network in Subsection 3.3 and present a method to train this model in Subsection 3.4;

### 3.1 Residual Scheduling

In our approach, the key is to remove irrelevant information, particularly for operations, from states (including partial solutions). An important benefit from this is that we do not need to include all irrelevant information while training to minimize the makespan. Let us illustrate by the state for the partial solution $S_3$ at time $\tau_3 = 3$ in Figure 2 (d). All processing by $\tau_3$ are irrelevant to the remaining scheduling. Since operations $O_{1,1}$ and $O_{2,1}$ are both finished and irrelevant the rest of scheduling, they can be removed from the state of $S_3$. In addition, operation $O_{2,2}$ is dispatched at time 2 (before $\tau_3 = 3$) and its processing time is $T_{2,1}^{(op)} = 4$, so the operation is marked as *ongoing*. Thus, the operation can be modified to start at $\tau_3 = 3$ with a processing time $4 - (3 - 2)$. Thus, the modified state for $S_3$ do not contain both $O_{1,1}$ and $O_{2,1}$, and modify $O_{2,2}$ as above. Let us consider two more examples. For $S_4$, one more operation $O_{2,2}$ is dispatched and thus marked as ongoing, however, the time $\tau_4$ remains unchanged and no more operations are removed. In this case, the state is almost the same except for including one more ongoing operation $O_{2,2}$. Then, for $S_5$, two more operations $O_{3,1}$

147 and $O_{2,2}$ are removed and the ongoing operation $O_{1,2}$ changes its processing time to the remaining
148 time (5-3).

149 For residual scheduling, we also reset the dispatching time $\tau = 0$ for all states with partial solutions
150 modified as above, so we derive makespans which is also irrelevant to the earlier operations. Given
151 a scheduling policy $\pi$, $T_\pi^{(mksp)}(S)$ is defined to be the makespan derived from an episode starting
152 from states $S$ by following $\pi$, and $T_\pi^{(mksp)}(S, a)$ the makespan by taking action $a$ on $S$.

### 3.2 Residual Graph Representation

154 In this paper, our model design is based on graph neural network (GNN), and leverage GNN to
155 extract the scheduling decision from the relationship in graph. In this subsection, we present the
156 graph representation. Like many other researchers such as Park et al. (2021a), we formulate a partial
157 solution into a graph $\mathcal{G} = (\mathcal{V}, \mathcal{E})$, where $\mathcal{V}$ is a set of nodes and $\mathcal{E}$ is a set of edges. A node is either a
158 machine node $M$ or an operation node $O$. An edge connects two nodes to represent the relationship
159 between two nodes, basically including three kinds of edges, namely operation-to-operation ($O \rightarrow O$),
160 machine-to-operation ($M \rightarrow O$) and operation-to-machine ($O \rightarrow M$). All operations in the same
161 job are fully connected as $O \rightarrow O$ edges. If an operation $O$ is able to be performed on a machine
162 $M$, there exists both $O \rightarrow M$ and $M \rightarrow O$ directed edges. In (Park et al., 2021a), they also let all
163 machines be fully connected as $M \rightarrow M$ edges. However, our experiments in section 4 show that
164 mutual $M \rightarrow M$ edges do not help much based on our Residual Scheduling. An illustration for graph
165 representation of $S_3$ is depicted in Figure 3 (a).

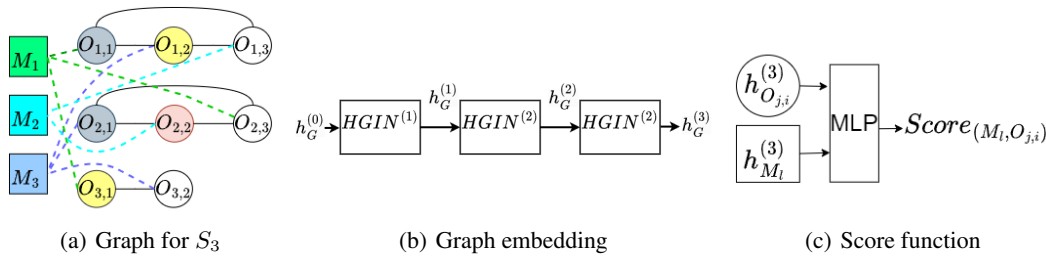

(a) Graph for $S_3$        (b) Graph embedding        (c) Score function

Figure 3: Graph representation and networks.

166 In the graph representation, all nodes need to include some attributes so that a partial solution $S$ at
167 the dispatching time $\tau$ can be supported in the MDP formulation (in the appendix). Note that many of
168 the attributes below are normalized to reduce variance. For nodes corresponding to operations $O_{j,i}$,
169 we have the following attributes:

170 ***Status*** $\phi_{j,i}$: The operation status $\phi_{j,i}$ is *completed* if the operation has been finished by $\tau$, *ongoing* if
171 the operation is ongoing (i.e., has been dispatched to some machine by $\tau$ and is still being processed
172 at $\tau$), *ready* if the operation designated to the machine which is idle has not been dispatched yet and
173 its precedent operation has been finished, and *unready* otherwise. For example, in Figure 3 (a), the
174 gray nodes are *completed*, the red *ongoing*, the yellow *ready* and the white *unready*. In our residual
175 scheduling, there exists no completed operations in all partial solutions, since they are removes for
176 irrelevance of the rest of scheduling. The attribute is a one-hot vector to represent the current status
177 of the operation, which is one of *ongoing*, *ready* and *unready*. Illustration for all states $S_0$ to $S_8$ are
178 shown in the appendix.

179 ***Normalized processing time*** $\bar{T}_{j,i}^{(op)}$: Let the maximal processing time be $T_{max}^{(op)} = \max_{\forall j,i}(T_{j,i}^{(op)})$.
180 Then, $\bar{T}_{j,i}^{(op)} = T_{j,i}^{(op)}/T_{max}^{(op)}$. In our residual scheduling, the operations that have been finished are
181 removed in partial solutions and therefore their processing time can be ignored; the operations that
182 has not been dispatched yet still keep their processing times the same; the operations that are *ongoing*
183 change their processing times to the remaining times after the dispatching time $\tau_t$. As for FJSP, the
184 operations that has not been dispatched yet may have several processing times on different machines,
185 and thus we can simply choose the average of these processing times.

186 ***Normalized job remaining time*** $\bar{T}_{j,i}^{(job)}$: Let the rest of processing time for job $J_j$ be $T_{j,i}^{(job)} =$
187 $\sum_{\forall i' \geq i} T_{j,i'}^{(op)}$, and let the processing time for the whole job $j$ be $T_j^{(job)} = \sum_{\forall i'} T_{j,i'}^{(op)}$. In practice,
188 $T_j^{(job)}$ is replaced by the processing time for the original job $j$. Thus, $\bar{T}_{j,i}^{(job)} = T_{j,i}^{(job)}/T_j^{(job)}$. For
189 FJSP, since operations $O_{j,i}$ can be dispatched to different designated machines $M_l$, say with the
190 processing time $T_{j,i,l}^{(op)}$, we simply let $T_{j,i}^{(op)}$ be the average of $T_{j,i,l}^{(op)}$ for all $M_l$.

191 For machine nodes corresponding to machines $M_l$, we have the following attributes:

192 ***Machine status*** $\phi_l$: The machine status $\phi_l$ is *processing* if some operation has been dispatched to
193 and is being processed by $M_l$ at $\tau$, and *idle* otherwise (no operation is being processed at $\tau$). The
194 attribute is a one-hot vector to represent the current status, which is one of *processing* and *idle*.

195 ***Normalized operation processing time*** $\bar{T}_l^{(mac)}$: On the machine $M_l$, the processing time $T_l^{(mac)}$ is
196 $T_{j,i}^{(op)}$ (the same as the normalized processing time for node $O_{j,i}$) if the machine status is *processing*,
197 i.e., some ongoing operation $O_{j,i}$ is being processed but not finished yet, is zero if the machine status
198 is *idle*. Then, this attribute is normalized to $T_{max}^{(op)}$ and thus $\bar{T}_l^{(mac)} = T_l^{(mac)}/T_{max}^{(op)}$.

199 Now, consider edges in a residual scheduling graph. As described above, there exists three relationship
200 sets for edges, $O \rightarrow O$, $O \rightarrow M$ and $M \rightarrow O$. First, for the same job, say $J_j$, all of its operation
201 nodes for $O_{j,i}$ are fully connected. Note that for residual scheduling the operations finished by the
202 dispatching time $\tau$ are removed and thus have no edges to them. Second, a machine node for $M_l$ is
203 connected to an operation node for $O_{j,i}$, if the operation $O_{j,i}$ is designated to be processed on the
204 machine $M_l$, which forms two edges $O \rightarrow M$ and $M \rightarrow O$. Both contains the following attribute.

205 ***Normalized operation processing time*** $\bar{T}_{j,i,l}^{(edge)}$: The attribute is $\bar{T}_{j,i,l}^{(edge)} = T_{j,i}^{(op)}/T_{max}^{(op)}$. Here,
206 $T_{j,i}^{(op)} = T_{j,i,l}^{(op)}$ in the case of FJSP. If operation $O_{j,i}$ is ongoing (or being processed), $T_{j,i}^{(op)}$ is the
207 remaining time as described above.

## 3.3 Graph Neural Network

209 In this subsection, we present our model based on graph neural network (GNN). GNN are a family
210 of deep neural networks (Battaglia et al., 2018) that can learn representation of graph-structured
211 data, widely used in many applications (Lv et al., 2021; Zhou et al., 2020). A GNN aggregates
212 information from node itself and its neighboring nodes and then update the data itself, which allows
213 the GNN to capture the complex relationships within the data graph. For GNN, we choose *Graph*
214 *Isomorphism Network* (*GIN*), which was shown to have strong discriminative power (Xu et al., 2019)
215 and summarily reviewed as follows. Given a graph $\mathcal{G} = (\mathcal{V}, \mathcal{E})$ and $K$ GNN layers ($K$ iterations),
216 GIN performs the $k$-th iterations of updating feature embedding $h^{(k)}$ for each node $v \in \mathcal{V}$:

$$h_v^{(k)} = MLP^{(k)}((1 + \epsilon^{(k)})h_v^{(k-1)} + \sum_{u \in N_b(v)} h_u^{(k-1)}), \tag{1}$$

217 where $h_v^{(k)}$ is the embedding of node $v$ at the $k$-th layer, $\epsilon^{(k)}$ is an arbitrary number that can be
218 learned, and $N_b(v)$ is the neighbors of $v$ via edges in $\mathcal{E}$. Note that $h_v^{(0)}$ refers to its raw features for
219 input. $MLP^{(k)}$ is a *Multi-Layer Perceptron* (*MLP*) for the $k$-th layer with a batch normalization
220 (Ioffe and Szegedy, 2015).

221 Furthermore, we actually use *heterogeneous GIN*, also called *HGIN*, since there are two types of
222 nodes, machine and operation nodes, and three relations, $O \rightarrow O$, $O \rightarrow M$ and $M \rightarrow O$ in the
223 graph representation. Although we do not have cross machine relations $M \rightarrow M$ as described above,
224 updating machine nodes requires to include the update from itself as in (1), that is, there is also one
225 more relation $M \rightarrow M$. Thus, HGIN encodes graph information between all relations by using the
226 four MLPs as follows,

$$h_v^{(k+1)} = \sum_{\mathcal{R}} MLP_{\mathcal{R}}^{(k+1)}((1 + \epsilon_{\mathcal{R}}^{(k+1)})h_v^{(k)} + \sum_{u \in N_{\mathcal{R}}(v)} h_u^{(k)}) \tag{2}$$

227 where $\mathcal{R}$ is one of the above four relations and $MLP_{\mathcal{R}}^{(k)}$ is the MLP for $\mathcal{R}$. For example, for $S_0$ in
228 Figure 2 (a), the embedding of $M_1$ in the $(k+1)$-st iteration can be derived as follows.

$$h_{M_1}^{(k+1)} = MLP_{MM}^{(k+1)}((1 + \epsilon_{MM}^{(k+1)})h_{M_1}^{(k)}) + MLP_{OM}^{(k+1)}(h_{O_{1,1}}^{(k)} + h_{O_{1,2}}^{(k)} + h_{O_{1,3}}^{(k)}) \tag{3}$$

Similarly, the embedding of $O_{1,1}$ in the $(k+1)$-st iteration is:

$$h_{O_{1,1}}^{(k+1)} = MLP_{OO}^{(k+1)}((1 + \epsilon_{OO}^{(k+1)})h_{O_{1,1}}^{(k)} + h_{O_{1,2}}^{(k)} + h_{O_{1,3}}^{(k)}) + MLP_{MO}^{(k+1)}(h_{M_1}^{(k)}) \tag{4}$$

In our approach, an action includes the two phases, graph embedding phase and action selection phase. Let $h_{\mathcal{G}}^{(k)}$ denote the whole embedding of the graphs $\mathcal{G}$, a summation of the embeddings of all nodes, $h_v^{(k+1)}$. In the graph embedding phase, we use an HGIN to encode node and graph embeddings as described above. An example with three HGIN layers is illustrated in Figure 3 (b).

In the action selection phase, we select an action based on a policy, after node and graph embedding are encoded in the graph embedding phase. The policy is described as follows. First, collect all *ready* operations $O$ to be dispatched to machines $M$. Then, for all pairs $(M, O)$, feed their node embeddings $(h_M^{(k)}, h_O^{(k)})$ into a MLP $Score(M, O)$ to calculate their scores as shown in Figure 3 (c). The probability of selecting $(M, O)$ is calculated based on a softmax function of all scores, which also serves as the model policy $\pi$ for the current state.

### 3.4 Policy-Based RL Training

In this paper, we propose to use a policy-based RL training mechanism that follows REINFORCE (Sutton and Barto, 2018) to update our model by policy gradient with a normalized advantage makespan with respect to a baseline policy $\pi_b$ as follows.

$$A_\pi(S, a) = \frac{T_{\pi_b}^{(mksp)}(S, a) - T_\pi^{(mksp)}(S, a)}{T_{\pi_b}^{(mksp)}(S, a)} \tag{5}$$

In this paper, we choose a lightweight PDR, MWKR, as baseline $\pi_b$, which performed best for makespan among all PDRs reported from the previous work (Zhang et al., 2020). In fact, our experiment also shows that using MWKR is better than the other PDRs shown in the appendix. The model for policy $\pi$ is parametrized by $\theta$, which is updated by $\nabla_\theta log\pi_\theta A_{\pi_\theta}(S_t, a_t)$. Our algorithm based on REINFORCE is listed in the appendix.

## 4 Experiments

### 4.1 Experimental Settings and Evaluation Benchmarks

In our experiments, the settings of our model are described as follows. All embedding and hidden vectors in our model have a dimension of 256. The model contains three HGIN layers for graph embedding, and an MLP for the score function, as shown in Figure 3 (b) and (c). All MLP networks including those in HGIN and for score contain two hidden layers. The parameters of our model, such as MLP, generally follow the default settings in PyTorch (Paszke et al., 2019) and PyTorch Geometric (Fey and Lenssen, 2019). More settings are in the appendix.

Each of our models is trained with one million episodes, each with one scheduling instance. Each instance is generated by following the procedure which is used to generate the TA dataset (Taillard, 1993). Given $(N, M)$, we use the procedure to generate an $n \times m$ JSP instance by conforming to the following distribution, $n \sim \mathcal{U}(3, N)$, $m \sim \mathcal{U}(3, n)$, and operation count $k_j = m$, where $\mathcal{U}(x, y)$ represents a distribution that uniformly samples an integer in a close interval $[x, y]$ at random. The details of designation for machines and processing times refer to (Taillard, 1993) and thus are omitted here. We choose (10,10) for all experiments, since (10,10) generally performs better than the other two as described in the appendix. Following the method described in Subsection 3.4, the model is updated from the above randomly generated instances. For testing our models for JSP and FJSP, seven JSP open benchmarks and two FJSP open benchmarks are used, as listed in the appendix.

The performance for a given policy method $\pi$ on an instance is measured by the makespan gap $G$ defined as

$$G = \frac{T_\pi^{(mksp)} - T_{\pi*}^{(mksp)}}{T_{\pi*}^{(mksp)}} \tag{6}$$

where $T_{\pi*}^{(mksp)}$ is the optimal makespan or the best-effort makespan, from a mathematical optimization tool, OR-Tools, serving as $\pi*$. By the best-effort makespan, we mean the makespan derived with a

Table 2: Average makespan gaps for TA benchmarks.

| Size | 15×15 | 20×15 | 20×20 | 30×15 | 30×20 | 50×15 | 50×20 | 100×20 | *Avg.* |
|------|-------|-------|-------|-------|-------|-------|-------|--------|--------|
| RS | 0.148 | **0.165** | 0.169 | **0.144** | **0.177** | **0.067** | **0.100** | **0.026** | **0.125** |
| RS+op | **0.143** | 0.193 | **0.159** | 0.192 | 0.213 | 0.123 | 0.126 | 0.050 | 0.150 |
| MWKR | 0.191 | 0.233 | 0.218 | 0.239 | 0.251 | 0.168 | 0.179 | 0.083 | 0.195 |
| MOR | 0.205 | 0.235 | 0.217 | 0.228 | 0.249 | 0.173 | 0.176 | 0.091 | 0.197 |
| SPT | 0.258 | 0.328 | 0.277 | 0.352 | 0.344 | 0.241 | 0.255 | 0.144 | 0.275 |
| FIFO | 0.239 | 0.314 | 0.273 | 0.311 | 0.311 | 0.206 | 0.239 | 0.135 | 0.254 |
| L2D | 0.259 | 0.300 | 0.316 | 0.329 | 0.336 | 0.223 | 0.265 | 0.136 | 0.270 |
| Park | 0.201 | 0.249 | 0.292 | 0.246 | 0.319 | 0.159 | 0.212 | 0.092 | 0.221 |
| SchN | 0.152 | 0.194 | 0.172 | 0.190 | 0.237 | 0.138 | 0.135 | 0.066 | 0.161 |

sufficiently large time limitation, namely half a day with OR-Tools. For comparison in experiments, we use a server with Intel Xeon E5-2683 CPU and a single NVIDIA GeForce GTX 1080 Ti GPU. Our method uses a CPU thread and a GPU to train and evaluate, while OR-Tools uses eight threads to find the solution.

## 4.2 Experiments for JSP

For JSP, we first train a model based on residual scheduling, named RS. For ablation testing, we also train a model, named RS+op, by following the same training method but without removing irrelevant operations. When using these models to solve testing instances, action selection is based on the greedy policy that simply chooses the action $(M, O)$ with the highest score deterministically, obtained from the score network as in Figure 3 (c).

For comparison, we consider the three DRL construction heuristics, respectively developed in (Zhang et al., 2020) called L2D, (Park et al., 2021b) by Park et al., and (Park et al., 2021a), called ScheduleNet. We directly use the performance results of these methods for open benchmarks from their articles. For simplicity, they are named L2D, Park and SchN respectively in this paper. We also include some construction heuristics based PDR, such as MWKR, MOR, SPT and FIFO. Besides, to derive the gaps to the optimum in all cases, OR-Tools serve as $\pi*$ as described in (6).

Now, let us analyze the performances of RS as follows. Table 2 shows the average makespan gaps for each collection of JSP TA benchmarks with sizes, 15×15, 20×15, 20×20, 30×15, 30×20, 50×15, 50×20 and 100×20, where the best performances (the smallest gaps) are marked in bold. In general, RS performs the best, and generally outperforms the other methods for all collections by large margins, except for that it has slightly higher gaps than RS+op for the two collections, $15 \times 15$ and $20 \times 20$. In fact, RS+op also generally outperforms the rest of methods, except for that it is very close to SchN for two collections. For the other six open benchmarks, ABZ, FT, ORB, YN, SWV and LA, the performances are similar and thus presented in the appendix. It is concluded that RS generally performs better than other construction heuristics by large margins.

## 4.3 Experiments for FJSP

Table 3: Average makespan gaps for FJSP open benchmarks

| Method | MK | LA(rdata) | LA(edata) | LA(vdata) |
|--------|-----|-----------|-----------|-----------|
| RS | **0.232** | **0.099** | **0.146** | 0.031 |
| RS+op | 0.254 | 0.113 | 0.168 | **0.029** |
| DRL-G | 0.254 | 0.111 | 0.150 | 0.040 |
| MWKR | 0.282 | 0.125 | 0.149 | 0.051 |
| MOR | 0.296 | 0.147 | 0.179 | 0.061 |
| SPT | 0.457 | 0.277 | 0.262 | 0.182 |
| FIFO | 0.307 | 0.166 | 0.220 | 0.075 |

For FJSP, we also train a model based on residual scheduling, named RS, and a ablation version, named RS+op, without removing irrelevant operations. We compares ours with one DRL construction heuristics developed by (Song et al., 2023), called DRL-G, and four PDR-based heuristics, MOR,

300 MWKR, SPT and FIFO. We directly use the performance results of these methods for open datasets
301 according to the reports from (Song et al., 2023).

302 Table 3 shows the average makespan gaps in the four open benchmarks, MK, LA(rdata), LA(edata)
303 and LA(vdata). From the table, RS generally outperforms all the other methods for all benchmarks
304 by large margins, except for that RS+op is slightly better for the benchmark LA(vdata).

## 5 Discussions

306 In this paper, we propose a new approach, called residual scheduling, to solving JSP an FJSP problems,
307 and the experiments show that our approach reaches SOTA among DRL-based construction heuristics
308 on the above open JSP and FJSP benchmarks. We further discusses three issues: large instances,
309 computation times and further improvement.

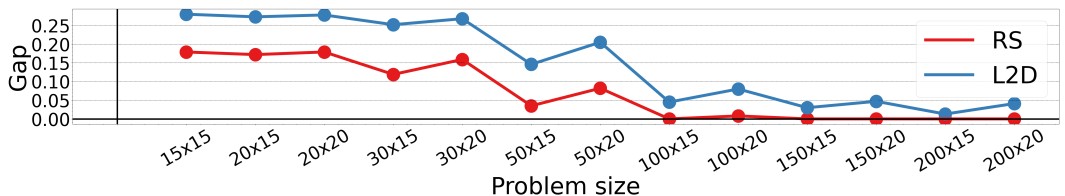

Figure 4: Average makespan gaps of JSP instances with different problem sizes.

310 First, from the above experiments particularly for TA benchmark for JSP, we observe that the average
311 gaps gets smaller as the number of jobs increases, even if we use the same model trained with
312 $(N, M) = (10, 10)$. In order to investigate size-agnostics, we further generate 13 collections of JSP
313 instances of sizes for testing, from $15 \times 15$ to $200 \times 20$, and generate 10 instances for each collection
314 by using the procedure above. Figure 4 shows the average gaps for these collections for RS and L2D,
315 and these collections are listed in the order of sizes in the x-axis. Note that we only show the results
316 of L2D in addition to our RS, since L2D is the only open-source among the above DRL heuristics.
317 Interestingly, using RS, the average gaps are nearly zero for the collections with sizes larger than 100
318 $\times$ 15, namely, $100 \times 15$, $100 \times 20$, $150 \times 15$, $200 \times 15$ and $200 \times 20$. Among the 50 JSP instances
319 in the five collections, 49 reaches zero gaps. A strong implication is that our RS approach can be
320 scaled up for job sizes and even reach the optimal for sufficient large job count.

321 Second, the computation times for RS are relatively small and has low variance like most of other
322 construction heuristics. Here, we just use the collection of TA 100x20 for illustration. It takes about
323 30 seconds on average for both RS and RS+op, about 28 for L2D and about 444 for SchN. In contrast,
324 it takes about 4000 seconds with high variance for OR-tools. The times for other collections are listed
325 in more detail in the appendix.

Table 4: Average makespan gaps for FJSP open benchmark.

| Method | MK | LA(rdata) | LA(edata) | LA(vdata) |
|---|---|---|---|---|
| RS | 0.232 | 0.099 | 0.146 | 0.031 |
| RS+100 | **0.154** | **0.047** | **0.079** | **0.007** |
| DRL-G | 0.254 | 0.111 | 0.150 | 0.040 |
| DRL+100 | 0.190 | 0.058 | 0.082 | 0.014 |

326 Third, as proposed by Song et al. (2023), construction heuristics can further improve the gap by
327 constructing multiple solutions based on the softmax policy, in addition to the greedy policy. They
328 had a version constructing 100 solutions for FJSP, called DRL+100 in this paper. In this paper, we
329 also implement a RS version for FJSP based on the softmax policy, as described in Subsection 3.3,
330 and then use the version, called RS+100, to constructing 100 solutions. In Table 4, the experimental
331 results show that RS+100 performs the best, much better than RS, DRL-G and DRL+100. An
332 important property for such an improvement is that constructing multiple solutions can be done in
333 parallel. That is, for construction heuristics, the solution quality can be improved by adding more
334 computation powers.

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
