# OpenReview forum: "Residual Scheduling: A New Reinforcement Learning Approach to Solving Job Shop Scheduling Problem"
_NeurIPS.cc/2023/Conference — Submitted to NeurIPS 2023_

### Official Review · Reviewer_xp5N · 2023-06-27

**Soundness:** 2 fair
**Presentation:** 2 fair
**Contribution:** 2 fair
**Rating:** 5
**Confidence:** 4

**Summary:**

This paper studied the problem of learning a graph neural network based policy network as a construction heuristic for solving job shop scheduling problems. This paper proposed an idea called residual scheduling to remove irrelevant operations and machines from the graph based state representation. This has been shown experimentally to perform well on several benchmark job shop scheduling problem instances.

**Strengths:**

With the fast advancement of deep learning technologies, it becomes increasingly interesting to develop deep neural network models that can effectively solve complex combinatorial optimization problems, including job shop scheduling problems. The newly developed deep learning system in this paper appears to be very effective and highly competitive in performance, compared to similar approaches from the literature.

**Weaknesses:**

The idea of using graph neural networks or other forms of deep neural networks trained through reinforcement learning to solve job shop scheduling problems has been studied in many past research works. The main text of this paper lacks a comprehensive review of these research works, making it hard to clearly understand the key technical novelty and contribution of this paper, compared to other recently published works.

The development of the residual scheduling technique is not strongly motivated in this paper. It is hard to understand why it is essential to remove irrelevant operations and machines from the graph based state representation. The development of this technique also appears to be very highly intuitive and lacks thorough theoretical analysis. Hence, the technical contribution of this development remains largely questionable.

The authors stated that in order for the process of learning the construction heuristic to be formulated as an MDP, they introduced several attributes for the operation nodes and machine nodes in the graph based state representation. However, it remains largely unknown whether, with the introduced attributes, the state representation can satisfy the Markov property of the MDP. The importance of using any newly introduced attributes should be more thoroughly evaluated experimentally. The associated technical contributions should be clarified and strongly justified.

Some other key aspects of the new system design should also be justified more. For example, the use of GIN needs to be supported with more convincing reasons. The focus on learning a construction heuristic rather than an improvement heuristic, which is gaining increasing popularity and attention, should be better justified and experimentally validated in this paper.

**Questions:**

What are the key novelties of this research work, compared to existing deep learning based methods for combinatorial optimization, including job shop scheduling?

At the theoretical level, why is it essential to remove irrelevant operations and machines from the graph based state representation?

With the newly introduced attributes, why will the state representation satisfy the Markov property of the MDP?

What are the theoretical and empirical advantages of learning a construction heuristic, in comparison to learning an improvement heuristic?

**Limitations:**

I do not have any concerns regarding this question.

---

> ### Author Rebuttal · Authors · 2023-08-09
>
> **Weaknesses**
>
> >The idea of using graph neural networks or other forms of deep neural networks trained through reinforcement learning to solve job shop scheduling problems has been studied in many past research works. The main text of this paper lacks a comprehensive review of these research works, making it hard to clearly understand the key technical novelty and contribution of this paper, compared to other recently published works.
>
> Due to page limitation, our review of previous works is in the Introduction section (L48 to L71) and briefly introduces our contribution in L72 to 75.
>
> >The development of the residual scheduling technique is not strongly motivated in this paper. It is hard to understand why it is essential to remove irrelevant operations and machines from the graph based state representation. The development of this technique also appears to be very highly intuitive and lacks thorough theoretical analysis. Hence, the technical contribution of this development remains largely questionable.
>
> About the contribution, please read those in the section of “Author Rebuttal by Authors” above.
>
> >The authors stated that in order for the process of learning the construction heuristic to be formulated as an MDP, they introduced several attributes for the operation nodes and machine nodes in the graph based state representation. However, it remains largely unknown whether, with the introduced attributes, the state representation can satisfy the Markov property of the MDP. The importance of using any newly introduced attributes should be more thoroughly evaluated experimentally. The associated technical contributions should be clarified and strongly justified.
>
> See the reply to the Question section.
>
> >Some other key aspects of the new system design should also be justified more. For example, the use of GIN needs to be supported with more convincing reasons. The focus on learning a construction heuristic rather than an improvement heuristic, which is gaining increasing popularity and attention, should be better justified and experimentally validated in this paper.
>
> See the reply to the Question section.
>
> **Questions**
>
> >What are the key novelties of this research work, compared to existing deep learning based methods for combinatorial optimization, including job shop scheduling?
>
> The comparison to existing work is described in the section of “Author Rebuttal by Authors” above.
>
> >At the theoretical level, why is it essential to remove irrelevant operations and machines from the graph based state representation?
>
> This issue is described in the section of “Author Rebuttal by Authors” above.
>
> >With the newly introduced attributes, why will the state representation satisfy the Markov property of the MDP?
>
> In this paper, the state representation satisfies the Markov property (of the MDP), like other works, such as L2D and ScheduleNet. Like many articles (e.g., Sutton's) said, to satisfy the Markov property, (1) State transition probabilities do not depend on the past state and action, and (2) Rewards do not depend on the past state and action.
>
> Our state representation satisfies the Markov property for the following:
>
> 1) State transition probabilities do not depend on the past state and action: For each state s, the transition to the next feasible state s' depends solely on the current operation node features and machine node features (note that when calculating the action selection probability, we only use the node features). This does not include information from earlier states, hence the probability of state transitions is independent of past states and actions. And the removal of nodes does not affect the above explanation.
>
> 2) Rewards do not depend on the past state and action: For any state s, the immediate reward for transitioning from state s to the next state is the increase in makespan after executing the selected action. This design ensures that the reward depends only on the current state and the selected action, and is independent of the earlier states and actions.

---

> ### Comment · Reviewer_xp5N · 2023-08-11
> **Thank the authors for their response**
>
> I would like to thank the authors for taking time to respond to my comments. Based on the response, I think my main concerns have not been properly addressed. The discussion on the Markov property did not seem to provide any additional insights on the system design and its novelty. The comparison to improvement heuristics both theoretically and empirically was not addressed with sufficient depth and details.

---

> > ### Author Response · Authors · 2023-08-12
> >
> > >I would like to thank the authors for taking time to respond to my comments. Based on the response, I think my main concerns have not been properly addressed. The discussion on the Markov property did not seem to provide any additional insights on the system design and its novelty.
> >
> > Thank you for your inquiry. In this paper, we simply said that for the approach of using solution constructions the state representation satisfies the Markov property (of the MDP). Some other works, such as L2D and ScheduleNet are also based on this approach (called construction heuristics). We did not claim novelty on this (similar to ScheduleNet), so we simply put it into the Appendix. We claim novelty and significance on "Residual Scheduling” as described in the Section of “Author Rebuttal by Authors”. And, we also claim that ours performed best among all works using construction heuristics.
> >
> > >The comparison to improvement heuristics both theoretically and empirically was not addressed with sufficient depth and details.
> >
> > ScheduleNet paper (Park et al 2021) has mentioned about the two types of heuristics as follows (Page 2 and 20): “the RL approaches can be categorized into: (1) improvement heuristics which learns to revise a complete solution iteratively to obtain a better solution; and (2) construction heuristics learns to construct a solution …. The improvement heuristics typically can obtain better performance than the construction heuristics as they find the best solution iteratively through the repetitive solution revising/searching process. However, improvement heuristics require expensive computations than construction heuristics.”
> >
> > So, this paper did not make more comparisons on this issue.
> >
> > To our knowledge, the improvement heuristics in L2S (Zhang et al. 2022) perform relatively well (probably the best) in the works of JSP with improvement heuristics. Here are some more points.
> > * Although improvement heuristics perform well for JSP with high steps (like 5000), these steps cannot be parallelized well. In contrast, most construction heuristics can construct solutions in parallel to further improve the performance. For example, our RS+100 for FJSP has much better performance than RS, and can be completely done in parallel.
> > In fact, we also have done experiments on RS+100 for JSP for comparisons to their works. The following table shows the performance results of RS+100 as well as others (including L2S-500, L2S-1000, L2S-2000, L2S-5000) for TA datasets. The results show that ours still outperforms L2S (even for L2S-5000) for large cases (50x15,50x20,100x20). This result also shows the superiority of our RS approach.
> >
> > |Size|15x15|20x15|20x20|30x15|30x20|50x15|50x20|100x20|Avg|
> > |-|-|-|-|-|-|-|-|-|-|
> > |RS|0.148|0.165|0.169|0.144|0.177|0.067|0.100|0.026|0.125|
> > |RS+100|0.109|0.111|0.117|0.108|0.141|**0.035**|**0.064**|**0.005**|0.086|
> > |L2S-500|0.093|0.116|0.124|0.147|0.175|0.110|0.130|0.079|0.122|
> > |L2S-1000|0.086|0.104|0.114|0.129|0.157|0.090|0.114|0.066|0.108|
> > |L2S-2000|0.071|0.094|0.102|0.110|0.140|0.069|0.093|0.051|0.091|
> > |L2S-5000|**0.062**|**0.083**|**0.090**|**0.090**|**0.126**|0.047|0.065|0.030|**0.074**|
> >
> > * L2S has been used for JSP so far. It is unclear whether it can be applied to FJSP well at least at this moment.

---

> > > ### Comment · Reviewer_xp5N · 2023-08-22
> > >
> > > Thank the authors for the further clarification. I have increased my rating a bit based on the additional information provided.

---

### Official Review · Reviewer_PVMB · 2023-06-30

**Soundness:** 3 good
**Presentation:** 3 good
**Contribution:** 3 good
**Rating:** 5
**Confidence:** 4

**Summary:**

The paper introduces a novel approach called residual scheduling for solving the Job-shop scheduling problem (JSP) and its variant, flexible JSP (FJSP), focusing on removing irrelevant machines and jobs from the consideration set. Despite these problems being NP-hard, the proposed method demonstrates state-of-the-art performance across standard benchmarks, even performing well when scaled to larger problem sizes, achieving a zero gap in 49 out of 50 instances with more than 150 jobs on 20 machines.

**Strengths:**

The method developed claims that for the 98% of the cases, the zero gap is achieved for fairly large instances.

**Weaknesses:**

The zero gap is mentioned. Upon reading, readers find out that the gap refers to "makespan gap." Understandably, significant bulk of existing papers on job-shop focus on makespan rather than tardiness. Ignoring tardiness may lead to poorer on time delivery, which is a weakness in itself, but with respect to the "makespan gap" measure it is not clear whether zero gap will result in zero (or small gap) is tardiness is considered or, generally, if the standard duality or MIP gaps are used instead. The gap is misleading at best.

**Questions:**

The paper claims to beath SOTA heuristics. Lagrangian heuristic is definitely missing. How does the new method compare against Lagrangian heuristic? Understandably, Lagrangian relaxation has been used for a long time, the reviewer is asking about recent rather than historical Lagrangian heuristics, since their quality may differ.

**Limitations:**

In light of the above comments, the limitations are 1. only makespan is considered, 2. only one type of gap seems to be introduced and considered, and 3. major heuristics are not used/compared with.

---

> ### Author Rebuttal · Authors · 2023-08-08
>
> **Weaknesses**
>
> >The zero gap is mentioned. Upon reading, readers find out that the gap refers to "makespan gap." Understandably, significant bulk of existing papers on job-shop focus on makespan rather than tardiness. Ignoring tardiness may lead to poorer on time delivery, which is a weakness in itself, but with respect to the "makespan gap" measure it is not clear whether zero gap will result in zero (or small gap) is tardiness is considered or, generally, if the standard duality or MIP gaps are used instead. The gap is misleading at best.
>
> To prevent misunderstanding, we will use “makespan gap” instead of “gap”.
>
> Since the majority of research in JSP/FJSP focuses on minimizing makespan as the objective, we also choose to optimize this objective to facilitate the comparison with other studies. The calculation of the gap is explicitly defined in equation (6).
>
> Besides, in our experiments as Figure 4 shown, RS obtains optimal solutions for all instances with sizes larger than 100x15. All details of optimal solutions are listed in Appendix Table 16 and Table 17.
>
> **Questions**
>
> >The paper claims to beat SOTA heuristics. Lagrangian heuristic is definitely missing. How does the new method compare against Lagrangian heuristic? Understandably, Lagrangian relaxation has been used for a long time, the reviewer is asking about recent rather than historical Lagrangian heuristics, since their quality may differ.
>
> Our paper only claims to achieve SOTA for all construction heuristics, not for all heuristics. For some other heuristics (e.g., GA, Tabu, and Lagrangian heuristic etc.), although some of them achieved better gaps (or smaller gaps), these methods usually take a lot more time. For example, for [1] (see below), it takes 3700, 5000, 4200, 7500 seconds (about 1-2 hours) to inference TA25 (20x20), TA30 (30x15), TA40(30x20), TA50 (50x15) respectively which we can inference within 2 seconds for these instances.
>
> [1] Kotary, J., Fioretto, F., & Van Hentenryck, P. (2022, June). Fast approximations for job shop scheduling: A lagrangian dual deep learning method. In Proceedings of the AAAI Conference on Artificial Intelligence (Vol. 36, No. 7, pp. 7239-7246).

---

> > ### Comment · Reviewer_PVMB · 2023-08-20
> >
> > The reviewer continues to have reservations regarding the choice of makespan. While it is acknowledged (by both the reviewer and the authors) that a significant portion of existing job-shop literature centers on makespan rather than tardiness, this choice restricts practical applicability, especially concerning on-time delivery—a factor crucial for customer satisfaction. Note that the acknowledgment of this predominant focus on makespan is a mere observation and does not constitute a rigorous scientific discovery. On the other hand, the discourse on LR heuristics/methods has somewhat improved and the results presented are promising. The most the reviewer can do in light of these concerns is adjust their rating from "borderline reject" to "borderline accept".

---

> > > ### Author Response · Authors · 2023-08-21
> > >
> > > Thank you very much for your positive re-evaluation and your acknowledgment of our results for makespan. Yes, we totally agree that other objectives like tardiness (more like machine utilization, setup cost, energy consumption, etc. [1]) are also important for practical applications, especially for scheduling for manufacturing plants, which we are actually also working with. We worked on the research towards RS actually based on our observation and analysis on these practical applications, with the objectives like makespan as well as tardiness/machine utilization/setup cost. In this paper, we target makespan, simply because it would be easier to make comparisons with existing job-shop research works on makespan (including datasets and benchmarks for comparisons). Like John McCarthy said "Chess as the Drosophila (the Fruit Fly) of Artificial Intelligence" in 1990, we usually want to work on simplified work first (makespan only) for a more complicated topic (including tardiness, etc).  I hope the above answer addresses your concerns on the issue of “makespan” only. Thank you very much again.
> > >
> > > [1] Xiong, Hegen, et al. "A survey of job shop scheduling problem: The types and models." Computers & Operations Research (2022)

---

### Official Review · Reviewer_e9mT · 2023-07-02

**Soundness:** 3 good
**Presentation:** 2 fair
**Contribution:** 2 fair
**Rating:** 5
**Confidence:** 5

**Summary:**

This paper proposed DRL based method to learn dispatching polices for (flexible) job-shop scheduling problems (JSP/FJSP). The main idea is to remove the completed operations from the state embedding, which is called residual scheduling, so as to improve the representation accuracy. The DRL agent uses a graph representation, which is processed by a Graph Neural Network (GNN) architecture. Experiments on JSP and FJSP benchmarks show that the proposed residual scheduling scheme outperforms recent DRL baselines.

**Strengths:**

1. The idea of residual scheduling makes great sense and is interesting.

2. The method is generally applicable to both JSP and FJSP, which are important scheduling problems.

3. Good empirical performance, comparing to recent DRL based scheduling methods.

**Weaknesses:**

1. The main weakness is that the technical contribution is incremental. While the redisual scheduling idea is interesting and novel, a large part of the proposed method is similar to existing works. Specifically, the graph representation and heterogeneous graph neural network in Section 3.2 and 3.3 is similar to the heterogeneous graph and heterogeneous GNN in (Song et al., 2023). This is not mentioned in Section 3, and the differences between the proposed method and existing works are not discussed.

2. Some design choices need to be justified. Please see the below questions.

3. Empirical evaluation needs to be improved.

* It is unclear whether the models in other works (e.g. L2D, SchN, DRL-G) are retrained using the same dataset as the proposed method. If not, then directly comparing their performance (even on the same benchmark instances) is not fair due to different training data.

* The discussion for Figure 4 is not surprising. It is well known that for JSP, problems with larger $n/m$ ratios are easier to solve (Taillard 1993). That is why the gaps on large problems in Figure 4 are smaller. Actually this is true for most algorithms, and cannot be claimed as a major advantage of the proposed method.

* Training time is not reported.

4. The authors made several inappropriate statements in the paper, mainly in introduction. The authors should be more precise about the related concepts.

* In the first paragraph, it is better to describe JSP as a combinatorial optimization problem, instead of mathematical optimization.

* In the second paragraph, Constraint Satisfaction Problem (CSP) should be not stated as a type of mathematical optimization. In addition, Constraint Programming (CP) here is more suitable than CSP.

* In the fourth paragraph, the approaches that automate the design of heuristics should be hyperheuristics, instead of metaheuristics.

4. The language needs to be improved. Besides, there are quite a few grammar errors and typos.

**Questions:**

1. Why use fully connected edges to link the operations in the same job? How about using only precedence relationship as a directed edge?

2. The second graph below Figure 3, for flexible problems, how to define the ready status for an operation considering there could be multiple compatible machines?

3. According to Figure 3(b), the graph embedding $h_{\mathcal{G}}^{(k)}$ is not used in the action score prediction as in Figure 3(c). So what is $h_{\mathcal{G}}^{(k)}$ used for?

4. In the experiments, to generate number of jobs and machines, why use the uniform distribution $\mathcal{U}(3, N)$ and $\mathcal{U}(3,n)$ (should be $\mathcal{U}(3,M)$ for machines)? The value 3 seems arbitrary.

**Limitations:**

The authors did not give a particular discussion on the limitations. One limitation could be the computational efficiency. As shown in Table 13 in the appendix, the runtime increase rapidly with the number of jobs. This could affect both training and inference efficiency, and limit its applicability to larger problems.

---

> ### Author Rebuttal · Authors · 2023-08-08
>
> **Weaknesses**
>
> >The main weakness is that the technical contribution is incremental. While the redisual scheduling idea is interesting and novel, a large part of the proposed method is similar to existing works. Specifically, the graph representation and heterogeneous graph neural network in Section 3.2 and 3.3 is similar to the heterogeneous graph and heterogeneous GNN in (Song et al., 2023).
> This is not mentioned in Section 3, and the differences between the proposed method and existing works are not discussed.
>
> First, thanks for finding it interesting for the novelty of RS, which, to our knowledge, none of previous works with construction heuristic (including L2D  (Zhang et al., 2020) , ScheduleNet (Park et al., 2021a), Song et al. (2023), Park et al., (2021b)) considered residual state. The significance and importance of RS are described in the common reply section as above.
>
> Ours is different from Song et al. (2023) as described in the section of “Author Rebuttal by Authors” above.
>
> >It is unclear whether the models in other works (e.g. L2D, SchN, DRL-G) are retrained using the same dataset as the proposed method. If not, then directly comparing their performance (even on the same benchmark instances) is not fair due to different training data.
>
> In fact, many of previous works did not provide their training data and their open-source version (only L2D has open-source). However, most of them used the similar procedure for the generated training data, which is mentioned in the Section 4.1. In this paper, we still use the same procedure to generate our training data and directly use their records for comparisons. In addition to this, we also use public datasets like TA, MK, etc. for comparisons.
>
> >The discussion for Figure 4 is not surprising. It is well known that for JSP, problems with larger $n/m$ ratios are easier to solve (Taillard 1993). That is why the gaps on large problems in Figure 4 are smaller. Actually this is true for most algorithms, and cannot be claimed as a major advantage of the proposed method.
>
> Yes, it is understandable that large JSP cases are easier to be solved. However, Figure 4  mainly wants to show that RS is much better than L2D, and actually achieves the same makespan as OR-Tools does for those large JSP cases.
>
> >Training time is not reported.
>
> It takes about one day to train with 200,000 episodes. We will add it in the revision.
>
> >In the first paragraph, it is better to describe JSP as a combinatorial optimization problem, instead of mathematical optimization.
>
> We said this since many articles considered combinatorial optimization is a subfield of mathematical optimization. In the revision, we will rephrase it.
>
> >In the second paragraph, Constraint Satisfaction Problem (CSP) should be not stated as a type of mathematical optimization. In addition, Constraint Programming (CP) here is more suitable than CSP.
>
> We will rephrase it in the revision.
>
> >In the fourth paragraph, the approaches that automate the design of heuristics should be hyperheuristics, instead of metaheuristics.
>
> Thanks for pointing this out. We will rephrase it to “a generic approach to search within a search space of problem solutions”.
>
> **Questions**
>
> >Why use fully connected edges to link the operations in the same job? How about using only precedence relationship as a directed edge?
>
> With fully connected edges, the embedding of one operation node encompasses all the information of all operations of the same job, thus potentially representing the job embedding, e.g., including information about the rest of the operations to be processed.
>
> >The second graph below Figure 3, for flexible problems, how to define the ready status for an operation considering there could be multiple compatible machines?
>
> The ready status for an operation is on as long as there exists at least one available machine that can process the operation at the time. So, it is the same for multiple available machines.
>
> >According to Figure 3(b), the graph embedding $h_{\mathcal{g}}^{(k)}$ is not used in the action score prediction as in Figure 3(c). So what is $h_{\mathcal{g}}^{(k)}$ used for?
>
> The variable $h_{\mathcal{g}}^{(k)}$ is to represent all of the hidden embeddings of the graph, including all $h_{O\in\mathcal{g}}^{(k)}$ and $h_{M\in\mathcal{g}}^{(k)}$. We will add the definition in the revision.
>
> >In the experiments, to generate number of jobs and machines, why use the uniform distribution $\mathcal{U}(3, N)$ and $\mathcal{U}(3, n)$   (should be $\mathcal{U}(3, M)$  for machines)? The value 3 seems arbitrary.
>
> Since the number of jobs, $n$, is greater than the number of machines, $m$, in most datasets of past works, we let $m\sim\mathcal{U}(3, n)$ such that $m\leq n$.
>
> The reason for choosing 3 as a lower bound for the number of jobs or machines is simply because we think that it is most likely to be a trivial case of no more than 2 jobs and 2 machines.

---

> > ### Comment · Reviewer_e9mT · 2023-08-19
> >
> > Thanks for the response, which addressed my concern. I increased my score to 5.

---

> > > ### Author Response · Authors · 2023-08-19
> > >
> > > Thank you very much for your positive response towards acceptance. In the case of acceptance, we will revise this paper accordingly based on your valuable comments. Again, thank you, and we are confident that this paper is worthy of this conference.

---

### Official Review · Reviewer_8Fub · 2023-07-10

**Soundness:** 2 fair
**Presentation:** 2 fair
**Contribution:** 2 fair
**Rating:** 6
**Confidence:** 5

**Summary:**

This paper proposes a deep reinforcement learning-based constructive heuristic to solve the (Flexible) Job Shop Scheduling Problem. An instance of the problem is represented as a graph and fed into a Graph Neural Network-based model which outputs a score for each candidate (operation-machine) pair. The model is trained with the REINFORCE algorithm with as baseline a classic Priority Dispatching Rule-based heuristic. The novelty of the paper lies in the update of the state after each action: the graph is updated by removing the operations which have already been executed to focus on the most relevant information, which is the residual operations and remaining times of the ongoing ones. The proposed approach is experimentally shown to outperform RL-based constructive heuristics on classic JSSP and FJSSP benchmarks.

**Strengths:**

1. Sound and interesting idea of removing irrelevant information from the state
1. The paper is fairly clear and I appreciated the illustrations Fig 1-3.
1. The model was proposed for the JSSP and easily adapted to the Flexible JSSP
1. The approach outperforms deep RL-based construction heuristics on classic benchmarks

**Weaknesses:**

1. Limited novelty: the main contribution is the definition of the residual state at each step of the construction process by removing irrelevant operations and resetting the time reference. This seems to me an incremental improvement of the approach L2D [1], which already proposed a similar state graph representation and the use of the Graph Isomorphism Network architecture for the JSSP.
1. The proposed approach seems very specific to (F)JSSPs (state representation, baseline) and it's not clear what could be transferable to DRL heuristics for solving other optimization problems.
1. In the experiments, the presented non-learning-based baselines seem pretty weak: only greedy (see more in Questions)
1. Lots of English typos (I noted a few per page)

[1] C Zhang et al, Learning to Dispatch for Job Shop Scheduling via Deep Reinforcement Learning, Neurips 2020

**Questions:**

1. Where dos the attributes of edges (L205) appear in the GNN model (Sec 3.3)?
1. The paper claims: “Interestingly, using RS, the average gaps are nearly zero for the collections with sizes larger than 100 … A strong implication is that our RS approach can be scaled up for job sizes and even reach the optimal for sufficient large job count.” Do you have an idea of why the model would work better on unseen large instances versus instances of the same size as the training ones? Does OR-tools return the optimal solutions for these larger instances? Can it be that the quality of the reference solutions decreases and therefore the “optimality” gap becomes smaller?
1. Are there stronger non-learning-based baselines for the JSPP other than the greedy PDR heuristics presented in Table 2? To be able to appreciate the performance of the proposed approach, it would be useful and more convincing to compare it to the best heuristics for this problem, beyond simple greedy ones. For example maybe [2]?
1. Appendix, Algorithm 1, Line 6:  to compute this makespan at state s_t, given action a_t, do you do a rollout of the current policy \pi_{\theta} with the updated parameter \theta? This would mean doing a rollout until the end of both the baseline and current policy at each step of the trajectory? How long did the training take?
1. How is the average computation time computed? (Table 13) In particular, was the policy applied to each instance individually or were instances batched?
1. The discussion L322 about the time it takes for RS/L2D/ScheduleNet versus OR-Tools can be a bit misleading: OR-tools is an exact solver. If spends a lot of time proving the optimality of the solution. Probably it could return a good quality solution much faster if used as a heuristic. In addition it runs on CPUs and not GPUs therefore just comparing the time is not really fair.

[2] CY Zhang, A tabu search algorithm with a new neighborhood structure for the job shop scheduling problem, Computers & Operations Research, 2007

**Limitations:**

Not explicitly addressed by the paper.

---

> ### Author Rebuttal · Authors · 2023-08-08
>
> **Weaknesses**
>
> >Limited novelty: the main contribution is the definition of the residual state at each step of the construction process by removing irrelevant operations and resetting the time reference.
>
> Please see the section of “Author Rebuttal by Authors” above.
>
> >This seems to me an incremental improvement of the approach L2D [1], which already proposed a similar state graph representation and the use of the Graph Isomorphism Network architecture for the JSSP.
>
> To our best knowledge, none of previous works with construction heuristic (including L2D  (Zhang et al., 2020) , ScheduleNet (Park et al., 2021a), Song et al. (2023), Park et al., (2021b)) considered residual state (RS). In addition, ours is also different from L2D as described in the section of “Author Rebuttal by Authors” above.
>
> >The proposed approach seems very specific to (F)JSSPs (state representation, baseline) and it's not clear what could be transferable to DRL heuristics for solving other optimization problems.
>
> This paper focuses on (F)JSSPs. Based on this, we expect to extend it to other optimization problems in the future.
>
> **Questions**
>
> >Where does the attributes of edges (L205) appear in the GNN model (Sec 3.3)?
>
> The attributes of edges (L202~205) are concatenated to each $O\to M$ & $M\to O$ message passing, as mentioned in Equation (3), which hide the attributes in $h$. To clarify this, we will update the equation as follows.
>
> $h_{M_1}^{(k+1)}=MLP_{MM}^{(k+1)}((1+\epsilon)h_{M_1}^{(k)})+MLP_{OM}^{(k+1)}((h_{O_{1,1}}^{(k)}||\bar{T_{1,1,1}})+(h_{O_{2,3}}^{(k)}||\bar{T_{2,3,1}})+(h_{O_{3,1}}^{(k)}||\bar{T_{3,1,1}}))$
>
> >Do you have an idea of why the model would work better on unseen large instances versus instances of the same size as the training ones?
>
> Yes, from our observation, makespans become larger for large problem sizes, the gaps (normalized to the large makespans in the definition Equation (6)) also become smaller. The reason why the model would work better on unseen large instances is argued in the section of  “Author Rebuttal by Authors” above.
>
> >Does OR-tools return the optimal solutions for these larger instances?
>
> Given half-a-day computation, OR tools returned optimal solutions for most instances except for about 20 instances. Interestingly, optimal solutions were obtained for all instances with sizes larger than 100x15.
>
> >Can it be that the quality of the reference solutions decreases and therefore the “optimality” gap becomes smaller?
>
> Like said above, optimal solutions were obtained for all instances with sizes larger than 100x15, so we do not think this is the issue.
>
> >Are there stronger non-learning-based baselines for the JSPP other than the greedy PDR heuristics presented in Table 2? To be able to appreciate the performance of the proposed approach, it would be useful and more convincing to compare it to the best heuristics for this problem, beyond simple greedy ones. For example maybe [2]?
>
> For many non-learning methods, like [2] that you mentioned, while solving the problems with low gaps (like OR-Tools), they actually take a (unstably) long time to solve. For example, for the instances, swv11-swv15, our methods take 1-3 seconds on average, while the method [2] took about 1 hour (which is reported in [2]).
>
> [2] CY Zhang, A tabu search algorithm with a new neighborhood structure for the job shop scheduling problem, Computers & Operations Research, 2007.
>
> >Appendix, Algorithm 1, Line 6: to compute this makespan at state s_t, given action a_t, do you do a rollout of the current policy \pi_{\theta} with the updated parameter \theta? This would mean doing a rollout until the end of both the baseline and current policy at each step of the trajectory? How long did the training take?
>
> In line 5 & 6, we simply use a baseline (MWKR in most of our experiments) to rollout, not our (current) policy. (Note: MWKR is much faster than our policy, and our training dataset is (10,10).) It takes about one day to train with 200,000 episodes.
>
> >How is the average computation time computed? (Table 13) In particular, was the policy applied to each instance individually or were instances batched?
>
> The computation time is calculated for each individual instance. Then, all of these times are averaged.
>
> >The discussion L322 about the time it takes for RS/L2D/ScheduleNet versus OR-Tools can be a bit misleading: ......
> Probably it could return a good quality solution much faster if used as a heuristic.
> In addition it runs on CPUs and not GPUs therefore just comparing the time is not really fair.
>
> For this question, we reran our program with CPU and found that our version with CPU (with single thread) takes roughly twice of computation time for our version with GPU (as shown in the paper). The details are also shown below.
>
> |Size|15x15|20x15|20x20|30x15|30x20|50x15|50x20|100x20|
> |:-|-|-|-|-|-|-|-|-|
> |CPU time (s)|1.30|1.55|1.45|3.67|4.65|10.04|13.10|51.47|
> |GPU time (s)|0.47|0.83|0.91|1.93|2.21|5.3|6.96|27.32|
>
> Now, for fairness, we let OR-tools use one thread run within the same times ($T$) as above for each group of dataset (in the row of CPU time). In this way, the obtained makespans have the gaps as shown in the following table. From the table, ours clearly outperformed OR-tools’ by a large margin when limiting the running time to $T$ for OR-tools. Even for 2$T$ and 4$T$, ours also clearly outperformed OR-tools’ except for some small cases, like 15x15, 20x15. For some large instances like 100x20, 50x20, 50x15, the table shows that longer times do not help improve much.
>
> |Size|15x15|20x15|20x20|30x15|30x20|50x15|50x20|100x20|
> |:-|-|-|-|-|-|-|-|-|
> |makespan gap by OR-Tools ($T$) |0.159|0.245|0.229|0.297|0.312|0.207|0.251|0.143|
> |makespan gap by OR-Tools (2$T$) |0.121|0.214|0.202|0.267|0.292|0.199|0.248|0.143|
> |makespan gap by OR-Tools (4$T$) |0.094|0.162|0.171|0.226|0.263|0.189|0.241|0.143|
> |makespan gap by RS ($T$) |0.148|0.165|0.169|0.144|0.177|0.067|0.100|0.026|

---

> > ### Comment · Reviewer_8Fub · 2023-08-21
> >
> > I thank the authors for their precise answers. I increase my score to 6.

---

### Author Rebuttal · Authors · 2023-08-10

Dear all reviewers, we appreciate your valuable comments. We would like to address the common concerns raised by reviewers.

**Novelty and contribution.**

In RS, the model simply focuses on the remaining non-dispatched operations, so the problem size is getting smaller as the process goes. This potentially leads to a more accurate prediction, as illustrated as follows. For example, for a 20x10 instance at the beginning, the original problem size is 20x10, however, the size is reduced to 10x10 after scheduling half of the operations in RS. In RS, scheduling at this point is like scheduling an instance with size close to 10x10 (without extra information of 20x10). Since the problem size (about 10x10) has been trained as described in Section 4.1, our method is able to predict well at this point (presume that the model is well trained). In brief, removing irrelevant operations allows the model to focus on the most critical parts of the problem and capture essential patterns and features, leading to better prediction.

In contrast, for other scheduling methods (like L2D, ScheduleNet), their graphs still include information with size close to 20x10 at this point. For example, a node of an operation which has been finished still exists with status “assigned” in ScheduleNet, while in RS the node is completely removed. Note: obviously the current state of these methods includes a lot of extra information, which usually requires more training to predict well.

Our experiments also justify the expected improvements over other methods. This is one of the major contributions in this paper.

|Method|Problem|Connection of operation nodes|Machine nodes|Action|Model|
|:-|-|-|-|-|-|
|L2D|JSP|precedences |no|choose operation|homogeneous GIN|
|ScheduleNet|JSP/mTSP|fully connected|yes|choose machine-operation pair|homogeneous GAT|
|Song [1] |FJSP|precedences |yes|choose machine-operation pair|heterogeneous GAT|
|RS|JSP/FJSP|fully connected|yes|choose machine-operation pair|heterogeneous GIN|

**Similarity to existing works.**

Here, we list the differences in the table above and will add more discussion in the revised version.

*(To reviewer 8Fub)* For L2D, ours is different in the following senses.
In addition, ours is also different from L2D in the following senses:
1) Nodes: RS uses two kinds of nodes (operation nodes and machine nodes), while L2D only uses operation nodes (and uses edges to link those with the same machines). Thus, RS uses heterogeneous GIN for two kinds of nodes, while L2D uses homogeneous GIN.
Besides, RS and L2D use different features. Mentioned in Section 3.2, RS uses three features for operation nodes, two features for machine nodes, and one edge feature for machine-operation edge. L2D uses two features for operation nodes and no machine nodes.
2) Actions: In RS, we consider to choose the eligible machine-operation pairs, i.e., "assigning an operation to an available machine", while L2D only considers the eligible operations, i.e., "assigning an available operation" since they only consider JSSP and thus this straightforward scheme (of L2D) won’t work for FJSP.
3) Rewards/Returns: In RS, the reward/return is the normalized advantage makespane with respect to a baseline policy, while L2D’s reward is the quality difference between two states.

*(To reviewer e9mT)* For (Song et al., 2023)[1], ours is different in the following senses.
1) Nodes and features: RS and (Song et al., 2023) use two kinds of nodes (operation nodes and machine nodes), but (Song et al., 2023) uses additional dummy operation nodes (start and end nodes). RS fully connects operation nodes belonging to the same job, while (Song et al., 2023) uses precedences to link nodes.
2) RS uses heterogeneous GIN for two kinds of nodes, while (Song et al., 2023) uses heterogeneous GAT. We use GIN instead, since it is shown in (Xu et al., 2019)[2] that GIN has strong discriminative power.

[1] Wen Song, Xinyang Chen, Qiqiang Li, and Zhiguang Cao, Flexible Job-Shop Scheduling via Graph Neural Network and Deep Reinforcement Learning, IEEE Trans. Ind. Informatics 2023.

[2] Keyulu Xu, Weihua Hu, Jure Leskovec, and Stefanie Jegelka, How Powerful are Graph Neural Networks? ICLR 2019.

---

> ### Author Response · Authors · 2023-08-18
>
> Dear reviewers, thank you very much for your useful and valuable comments. We would appreciate your further comments and reconsidering the acceptance of this paper. We believe that this paper is worthy of acceptance as follows.
>
> The key contribution of this paper is that the proposed method RS for JSP/FJSP problems reaches State-of-the-Art (SoTA) among the methods with construction heuristics. RS leverages the idea of focusing on residual graph which is **DIFFERENT** from other methods, and benefit **the performance significantly particularly for large JSP/FJSP instances**.
>
> The advantage of RS (our method) becomes even clearer when compared to improvement heuristics (as also raised by reviewer xp5N). We quote from ScheduleNet paper (Park et al. 2021) which summarized the two types of heuristics for RL as follows (Page 20):
> > “the RL approaches can be categorized into: (1) improvement heuristics which learns to revise a complete solution iteratively to obtain a better solution; and (2) construction heuristics learns to construct a solution …. The improvement heuristics typically can obtain better performance than the construction heuristics as they find the best solution iteratively through the repetitive solution revising/searching process. However, improvement heuristics require expensive computations than construction heuristics.”
>
> To the best of our knowledge, L2S (Zhang et al. 2022) is the SoTA among all methods with the improvement heuristics. Based on the argument from the previous (Park et al. 2021), it is expected to have better solutions for improvement heuristics. But, even for that, our RS+100 (with construction heuristics) for JSP is competitive to L2S (with improvement heuristics) in terms of better solutions, regardless of lower computation costs. (Note: RS+100 can even reach the same computation time as RS while being processed in parallel.) Especially, our RS+100 outperforms all L2S (L2S-500 to L2S-5000) for large cases (50x15,50x20,100x20). These results (as also in the following table) also show the superiority of our RS approach.
> |Size|15x15|20x15|20x20|30x15|30x20|50x15|50x20|100x20|Avg|
> |-|-|-|-|-|-|-|-|-|-|
> |RS|0.148|0.165|0.169|0.144|0.177|0.067|0.100|0.026|0.125|
> |RS+100|0.109|0.111|0.117|0.108|0.141|**0.035**|**0.064**|**0.005**|0.086|
> |L2S-500|0.093|0.116|0.124|0.147|0.175|0.110|0.130|0.079|0.122|
> |L2S-1000|0.086|0.104|0.114|0.129|0.157|0.090|0.114|0.066|0.108|
> |L2S-2000|0.071|0.094|0.102|0.110|0.140|0.069|0.093|0.051|0.091|
> |L2S-5000|**0.062**|**0.083**|**0.090**|**0.090**|**0.126**|0.047|0.065|0.030|**0.074**|
>
> In short, we believe RS is an important contribution. Given that the author-reviewer discussion phase will end on Aug 21, we would be grateful if reviewers would kindly acknowledge our rebuttals, and appreciate your reconsideration. Thank you very much again.

---

> > ### Author Response · Authors · 2023-08-20
> >
> > Dear reviewers and chairs, we sincerely thank you very much for your valuable comments and discussions.
> > Given that the author/reviewer discussion phase will end very soon (on Aug 21), we would be very grateful if you would acknowledge our rebuttals. We want to ensure to have the chance to clarify questions or concerns during the phase.
> > Again, we appreciate your reconsideration on the acceptance of this paper, if not respond yet. We are confident that this paper is worthy of this conference.
> > While we know novelty is critical, it is sort-of subjective about how big novelty/contribution a work is.
> > (Note that a simple approach is often overlooked as limited or minor novelty.)
> > In contrast, it is objective about the performance.
> > This paper proposes Residual Scheduling which performs "consistently" well, especially for large instances (please read the previous message) for which RS clearly outperformed other RL methods with construction heuristics and even for those with improvement heuristics.
> > This shows the merit and importance of the "residual" concept, though simple, which was overlooked in many previous papers, some in top-tier conferences (including NeurIPS) and journals.
> > Finally, we again very much appreciate your comments and reconsideration with the support of acceptance.

---

### Decision · Program_Chairs · 2023-09-21

**Decision:**

Reject

**Comment:**

I thank the reviewers for their comments. I took the discussion as well as the author feedback into consideration. there seems to be general consensus among the reviewers that the paper is not quite there yet for NeurIPS but I encourage the authors to revise their work and taking the comments of the reviewers into account. In fact, as it stands we have only weak and borderline accepts and nobody is really championing the paper.